# Multisensor Assessment of Leaf Area Index across Ecoregions of Ardabil Province, Northwestern Iran

Lida Andalibi [1], Ardavan Ghorbani [2,*], Roshanak Darvishzadeh [3], Mehdi Moameri [2], Zeinab Hazbavi [2], Reza Jafari [4] and Farid Dadjou [1]

1   Department of Natural Resources, University of Mohaghegh Ardabili, Ardabil 56199-11367, Iran
2   Department of Natural Resources, Water Management Research Center, University of Mohaghegh Ardabili, Ardabil 56199-11367, Iran
3   Faculty of Geo-Information Science and Earth Observation (ITC), University of Twente, 7500 AE Enschede, The Netherlands
4   Department of Natural Resources, Isfahan University of Technology, Isfahan 84156-83111, Iran
*   Correspondence: a_ghorbani@uma.ac.ir; Tel.: +98-912-665-2624

**Abstract:** Leaf area index (LAI), one of the most crucial vegetation biophysical variables, is required to evaluate the structural characteristic of plant communities. This study, therefore, aimed to evaluate the LAI of ecoregions in Iran obtained using Sentinel-2B, Landsat 8 (OLI), MODIS, and AVHRR data in June and July 2020. A field survey was performed in different ecoregions throughout Ardabil Province during June and July 2020 under the satellite image dates. A Laipen LP 100 (LP 100) field-portable device was used to measure the LAI in 822 samples with different plant functional types (PFTs) of shrubs, bushes, and trees. The LAI was estimated using the SNAPv7.0.4 (Sentinel Application Platform) software for Sentinel-2B data and Google Earth Engine (GEE) system–based EVI for Landsat 8. At the same time, for MODIS and AVHRR, the LAI products of GEE were considered. The results of all satellite-based methods verified the LAI variations in space and time for every PFT. Based on Sentinel-2B, Landsat 8, MODIS, and AVHRR application, the minimum and maximum LAIs were respectively obtained at 0.14–1.78, 0.09–3.74, 0.82–4.69, and 0.35–2.73 for shrubs; 0.17–5.17, 0.3–2.3, 0.59–3.84, and 0.63–3.47 for bushes; and 0.3–4.4, 0.3–4.5, 0.7–4.3, and 0.5–3.3 for trees. These estimated values were lower than the LAI values of LP 100 (i.e., 0.4–4.10 for shrubs, 1.6–7.7 for bushes, and 3.1–6.8 for trees). A significant correlation ($p < 0.05$) for almost all studied PFTs between LP 100-LAI and estimated LAI from sensors was also observed in Sentinel-2B ($|r| > 0.63$ and $R^2 > 0.89$), Landsat 8 ($|r| > 0.50$ and $R^2 > 0.72$), MODIS ($|r| > 0.65$ and $R^2 > 0.88$), and AVHRR ($|r| > 0.59$ and $R^2 > 0.68$). Due to its high spatial resolution and relatively significant correlation with terrestrial data, Sentinel-2B was more suitable for calculating the LAI. The results obtained from this study can be used in future studies on sustainable rangeland management and conservation.

**Keywords:** condition monitoring; multiresolution; spatial analysis; spectral data; woody species

## 1. Introduction

Leaf area index (LAI) is defined as a one-sided leaf area per unit area of land [1,2]. Knowledge of the LAI is used in many disciplines, including investigation of stresses caused by environmental conditions, estimation of evapotranspiration, respiration, carbon and nutrient cycle, net primary production, and rainfall interception by woody plants [3]. Direct and indirect methods estimate the vegetation LAI in the field [4]. Direct methods are laborious and time-consuming; however, they provide reliable LAI estimations and are used to validate indirect methods [5]. A common method of remote-sensing-based (indirect) methods in estimating the LAI is to establish a physical relationship between the LAI and various vegetation indices extracted from visible near-infrared (VNIR) and short-wavelength infrared (SWIR) bands, including enhanced vegetation index (EVI), soil-adjusted vegetation index (SAVI), and normalized difference vegetation index (NDVI) [6].

These high multispectral- and hyperspectral-based indices significantly enhance the quality of monitoring the health of natural ecosystems and detecting the changes in vegetation biophysical characteristics [6,7].

Over the past decades, many efforts have focused on LAI estimation using ground-based field measurements (direct method) and remote sensing data (indirect method) [8,9]. Remote sensing methods have unique advantages in estimating the LAI over a large area. For example, the LAI obtained from optical remotely sensed data serves as a pivotal variable to estimate the aboveground biomass of forest stands [10]. Recently, indirect optical methods without contact with leaves, based on the radiation transmission and gap fraction theory, an array of commercial optical instruments, such as Plant Canopy Analyzer (LI-COR, Lincoln, NE, USA), DEMON (CISRO, Center for Environmental Mechanics, Canberra, Australia), Ceptometer (Decagon Device, Pullman, WA, USA), and digital camera with a fisheye lens, have been developed to estimate the effective LAI [6,7]. Due to the surface heterogeneity (different types of mixed coatings in pixel images) and temporal variability of plants in different growing seasons, the maximum accuracy of estimating the LAI by remote sensing data can only reach about 50% [8]. Therefore, it is necessary to increase the accuracy of estimating the LAI at different time–space scales. In estimating the LAI using remote sensing methods, it is always assumed that the leaves are homogeneously distributed, and the values obtained are known as the effective LAI [11].

In LAI values greater than 3, NDVI loses its sensitivity to changes in leaf green index or becomes saturated [2,12]. Therefore, in high LAI values, it is recommended to use the EVI instead the NDVI. The EVI affords thorough figures on spatial and temporal variations of vegetation, and it reduces the problems of impurities that the NDVI causes [13]. According to the remote sensing method in LAI estimation, vegetation includes all green factors, such as under forest canopies, including subfloors [7]. The LAI estimated using the Laipen LP 100 (LP 100) device is pure. After considering the type of plant under study and determining a coefficient in the obtained value, it becomes the effective LAI. Remote sensing observations are sensitive to the effective LAI [14]. The difference between the actual and effective LAI may be determined by the population index [15], which varies approximately between 0.5 (highly clustered canopies) and 1 (leaves with random distribution) [16].

In the remote sensing method, LAI retrieval has been achieved in medium-resolution spatial satellite images such as Sentinel-2 (with a resolution of 10 m [17]) in the SNAPv7.0.4 (Sentinel Application Platform) software, Landsat 8 OLI (with a resolution of 30 m) with the EVI in the Google Earth Engine (GEE) system [14], ready-made MODIS satellite products (with a resolution of 500 m; [18–20]), and ready product of AVHRR (with a resolution of 5566 m [21]). For Sentinel-2B, an operational LAI product associated with a quality indicator is provided through the SNAP toolbox and produced through a neural network trained by simulated spectra generated from well-known radiative transfer models (RTMs) [22]. Chen et al. [23] found that the estimated LAI in images with a larger scale has an error of about 25–50% due to surface heterogeneity. Liu et al. [24] also concluded that the LAI values obtained from MODIS images are consistently underestimated. Claverie et al. [21] used Sentinel-2B and Landsat 8 images to estimate the LAI and vegetation indices in the boreal forests of Finland. Their results showed that the obtained values were significantly different. They also showed the better performance of the Sentinel-2B image for the 705 nm red-edge bands. Brown et al. [25] estimated the LAI and chlorophyll content of vegetation using Sentinel-2 images. Chrysafis et al. [26] also concluded that the recovered LAI using Sentinel-2 images in a mixed Mediterranean forest region in Greece showed that the model obtained from the selection of spectral variables produced the most accurate LAI predictions with a coefficient of determination ($R^2$) of 0.85. Ovakoglou et al. [27] attempted to enhance the spatial resolution of the MODIS LAI product to the Landsat 8 resolution level. The estimated LAI values highly correlated with field-measured LAI during the dry period (0.72 < r < 0.94).

The coarse spatial resolution of satellite-based products does not authorize distinctive vegetation types within mixed pixels. Investigating only the prevailing type per pixel has

two major shortcomings: (a) the LAI of the prevailing vegetation type is contaminated by a spurious signal from other vegetation types, and (b) at the global scale, large regions of discrete vegetation types are ignored. To accurately estimate the LAI with remote sensing, spatial, temporal, and spectral resolutions at different scales must be carefully selected [9]. In this regard, the main purpose of this paper was to assess the performance of four sensors with different resolutions (i.e., Sentinel–2B (10 m), Landsat 8 (30 m), MODIS (500 m), and AVHRR (5566 m)) in estimating the LAI. LAI maps were plotted on all of these sensors as a reference for remote sensing data analysis with field data. This study provides a quantitative assessment of the quality of different sensors in estimating the LAI for the different scientific communities and software users.

## 2. Materials and Methods

### 2.1. Studied Area

Ardabil Province (1.80 million km$^2$) is located northwest of Iran (Figure 1). The studied area has a mountainous texture with high elevation differences, and the rest is plain and flat. The elevation varies from 20 to 4811 m above sea level (masl). Analysis of the Ardabil Province Meteorological Organization's statistics attributed the highest mean annual precipitation (between 400 and 500 mm) to Mt. Sabalan (western part of the province). A moderate value of mean annual precipitation is also observed for southern regions (350 mm), and its lowest allocated to the north of the province (210 to 240 mm). Moreover, the mean minimum and maximum temperatures are 1.50 and 20.50 °C, respectively [28–30]. Mostafazadeh and Mehri [31] also reported two main regimes for the precipitation seasonality of the province: a short dry season (seasonality index (SI) = 0.2–0.3) and a wet season (SI = 0.6). The rivers and water bodies of this province include Aras, Qarasu, Balkhlychay, Givichay, Shahrood, Qezelozen, Neur, and Shourabil lakes, which play a significant role in the formation of the climate and the moisture source of the province. Moreover, the Caspian Sea has a significant effect on its climatic regimes [16]. Given that the growing and the rainy season is from late March at low altitudes to mid-September at high elevations, the best time to collect data is June to July [32].

Some of the ecoregion covers contain geographically distinct sets of communities, natural species, and sub-ecoregions, which are dominated by three plant functional types (PFTs), including shrubs, bushes, and trees. A woody plant with a height of less than 50 cm and a size of small to medium is considered a shrub. Meanwhile, a woody plant with a height of 50–7 m and plenty of branches growing from both the ground and stiff stems is considered a bush. Finally, a woody plant with a height of more than 7 m, a single elongated stem, and few or no branches on its lower part is considered a tree. These PFTs were considered for the target ecoregions [14,32,33], distributed in Andabil, Bilesavar-Khoroslo, Darband Hir, Germi, Hashtjin, Hatam Meshasi forests, Khalkhal forests, Kowsar, Namin, and Neur Lake highlands (Figure 1).

### 2.2. Methodology

#### 2.2.1. Field Data Collection (LP 100 Device)

The field-based observations were recorded using the LP 100 device (Figure 2) during June and July 2020. The LP 100 device was applied to 160 shrubs, 117 bushes, and 455 trees throughout the studied ecoregions (Table 1). Therefore, in total, 822 ground truth points were sampled. Ecoregions that included only shrubs, bushes, and trees' dominance were selected (Figure 1) because of LP 100 limitation for LAI estimation for other PFTs (i.e., grasses, forbs, and dwarfs plant species). The LAI of the Hir, Neur, Kowsar, Meshginshahr, and Namin ecoregions were estimated in June 2020, and the others were studied in July 2020. The collected samples were representative of each ecoregion. All recorded values were transferred to the computer system. Then, the FluorPen software was applied to obtain the final LAI. It is noteworthy to note that the mean LAI of those PFTs located in one pixel was compared with the mean LAI estimated by different remote sensing methods of the same pixel.

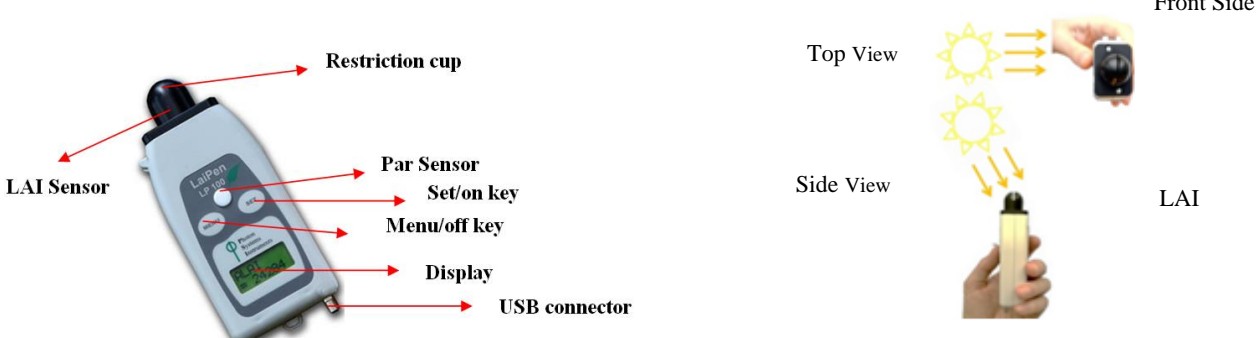

**Figure 1.** A general schematic of the location of Ardabil Province, its elevation, synoptic stations, and ground truth points.

**Figure 2.** Elements of LP 100 device (**left**) and its correct position for calculating ALAI (**right**) (Laipen LP 100 Manual, 2015).

**Table 1.** Sampling number for each plant functional type (PFT) at the studied ecoregions.

| PFTs | Number of Samples | Ecoregion (Sub-Ecoregion) | Sampling Month in 2020 |
|---|---|---|---|
| Shrubs | 13 | Andabil | July |
| | 13 | Hashtjin (Aghdagh, Berandagh) | July |
| | 28 | Khalkhal (Isbo, Jafarabad, Majareh, Dilmadeh, Shormineh, Chenarlagh) | July |
| | 26 | Kowsar (Mashkoul) | June |
| | 15 | Hatam Meshasi | June |
| | 65 | Namin Highlands | June |
| Sum | 160 samples | | |
| Bushes | 48 | Neur | June |
| | 9 | Bilesavar-Khoroslo | July |
| | 13 | Germi | July |
| | 13 | Andabil | July |
| | 10 | Hashtjin (Aghdagh, Berandagh) | July |
| | 10 | Khalkhal (Isbo, Jafarabad, Majareh, Dilmadeh, Shormineh, Chenarlagh) | July |
| | 5 | Hatam Meshasi | June |
| | 9 | Namin Highlands | June |
| Sum | 117 samples | | |
| Trees | 55 | Darband Hir | June |
| | 10 | Neur | June |
| | 16 | Germi | July |
| | 49 | Andabil | July |
| | 121 | Hashtjin (Aghdagh, Berandagh) | July |
| | 73 | Khalkhal (Isbo, Jafarabad, Majareh, Dilmadeh, Shormineh, Chenarlagh) | July |
| | 61 | Kowsar (Mashkoul) | June |
| | 70 | Hatam Meshasi | June |
| | 90 | Namin Highlands | June |
| Sum | 545 samples | | |

The ground sampling of three vegetation forms (shrubs, bushes, and trees) was conducted, taking into account the large pixel size of the selected sensors. Accordingly, the selected plants were first selected in large homogeneous areas; the size of these areas is much larger than the pixels of the selected sensors. Second, the studied PFTs were also tried to be selected at a distance from each other, so that they were representative of the average of the large area, which was selected. However, in general, the other vegetative forms (grasses and forbs) in the study area and the limitation of the device used (Laipen LP 100) are still a problem, which has not been possible to cover due to the limitations of this study.

Unlike other similar LAI measuring instruments, LP 100 is accurate in most daylight conditions, and there is no need to cover the cloud or a specific angle of the sun (Laipen LP 100 Manual [34]). Empirical law states the relationship between the intensity of light absorbed by the passage of homogeneous objects without scattering and the properties of matter modified by Monsi-Saeki. Therefore, the following equations are presented to correct the estimated radiation intensity under the vegetation canopy [35]:

$$I = I_0 e(-KLAI) \tag{1}$$

$$LAI = -Ln\left(\frac{I}{I_0}\right)/K \tag{2}$$

where $I$ is the radiation intensity in the lower part of the vegetation, and $I_0$ is the radiation intensity in the upper part of the vegetation. $K$ is the extinction factor depending on the vegetation canopy shape, orientation, and position. Given that the radiation intensity

decreases from top to bottom during its penetration, according to the Beer–Lambert law, it is necessary to use a correction factor to estimate the LAI of certain species [1].

As shown in Figure 2, the LP 100 device is placed in the plant's shade and under the leaf due to the measurement of the LAI under heterogeneous cover with direct sunlight. To estimate the LAI in each ecoregion, all PFTs were also considered. As the angle of view of the LAI sensor is open (112° on the horizontal axis), it is necessary to prevent direct light entering into the restriction cup. In other words, overexposure to an LAI sensor can misinterpret actual light conditions. Therefore, before each measurement, it is essential to place the device, as shown in Figure 2, to follow the standard principles. It is noteworthy that the results of LP 100 validation were already verified through the VitiCanopy app (r > 0.91; $R^2$ > 0.83; RMSE < 0.51) [14].

### 2.2.2. Image Selection and Image Preprocessing

The Sentinel-2B satellite has visible, near-infrared, and shortwave infrared (SWIR) sensors [36]. One of the advantages of using Sentinel-2B and Landsat 8 images is the high resolution of the images. In addition, the Sentinel-2B data are available for download from the European Space Agency Copernicus Open Access Hub (https://scihub.copernicus.eu/dhus/#/home; accessed on 28 November 2020). Using Sentinel-2B images with high revision time and images with a spatial resolution up to 10 m at no cost increases the accuracy of analysis of biophysical variables, such as the LAI.

The Sentinel-2B images were obtained from the Copernicus Open Access Hub (https://scihub.copernicus.eu; accessed on 28 November 2020) concurrent with the growth time of the PFTs and the field survey (June and July 2020, Table 2). At the same time, Landsat 8 images were taken from the USGS website and the GEE system. Landsat 8 images corrected in the GEE system are free [36]. Preprocessing was considered to extract the information from the images used accurately. To this end, atmospheric correction due to the effect of the atmosphere on the reflection of surface phenomena and its effect on the obtained result was considered [35]. Radiometric corrections also must be made to check for changes in the landscape, exposure, geometric visibility, weather conditions, and sensor noise [34].

**Table 2.** Selected satellite images and products.

| Satellite/Sensor | Date | Website/Products | |
|---|---|---|---|
| Sentinel-2B | | http://scihub.copernicus.eu (accessed on 28 November 2020) | Level-1C |
| Landsat 8 OLI | June–July 2020 | https://earthexplorer.usgs.gov/ (accessed on 28 November 2020) | - |
| MODIS * | | Terra + Aqua-4-Day L4Global 500 m | MCD15A3H |
| AVHRR | | (LAI_PAL_BU_V3) 5566 m | LAI_FAPAR/V5′ |

\* The LAI product has reached stage 2 validation. More details on MODIS land product validation for LAI/FPAR data products are available on the MODIS land team validation site. MCD15A3H Version 6 (MODIS) Level 4 Medium Resolution Imaging Spectrometer, Combined Fraction of Photosynthetic Active Radiation (FPAR), and LAI are a combined 4-day dataset of 500 m pixels.

For Sentinel-2B images, radiometric corrections, such as the calculation of radiance and atmospheric correction, were performed using the software. The Sen2Cor processor algorithm is a combination of the most advanced methods for modifying the Sentinel-2B atmosphere and comes with a module that fits the category. Images or products provided by GEE do not require preprocessing, such as geometric or radiometric correction, which is its advantage [36].

The SNAPv7.0.4 software was used to estimate the LAI from Sentinel-2B images after calling the images and atmospheric correction in the Sen2Cor plugin and then resampling it to an image with an accuracy of 10 m. Furthermore, for the Landsat 8 image, the EVI was selected to calculate the LAI in the GEE system. After calling the Landsat 8 images with

clouds less than 45% in Google Earth Engine, the EVI was calculated using the following formula [37]:

$$EVI = \frac{2.5 \times (NIR - RED)}{(NIR + 6.5 \times RED - 7.5 \times BLUE + 1)} \tag{3}$$

$$LAI = 3.618 \times EVI - 0.118 \tag{4}$$

where EVI, NIR, RED, and BLUE are respectively indicated as the enhanced vegetation index, near-infrared band, red band, and blue band.

The algorithm selects the best available pixels from the four MODIS sensors on NASA's Terra and Aqua satellites every 4 days. To convert the MODIS data scale, a correction factor of 0.1 was considered according to the information in its product (https://modis-land.gsfc.nasa.gov; accessed on 1 December 2020).

The methodological flowchart of the current research is shown in Figure 3.

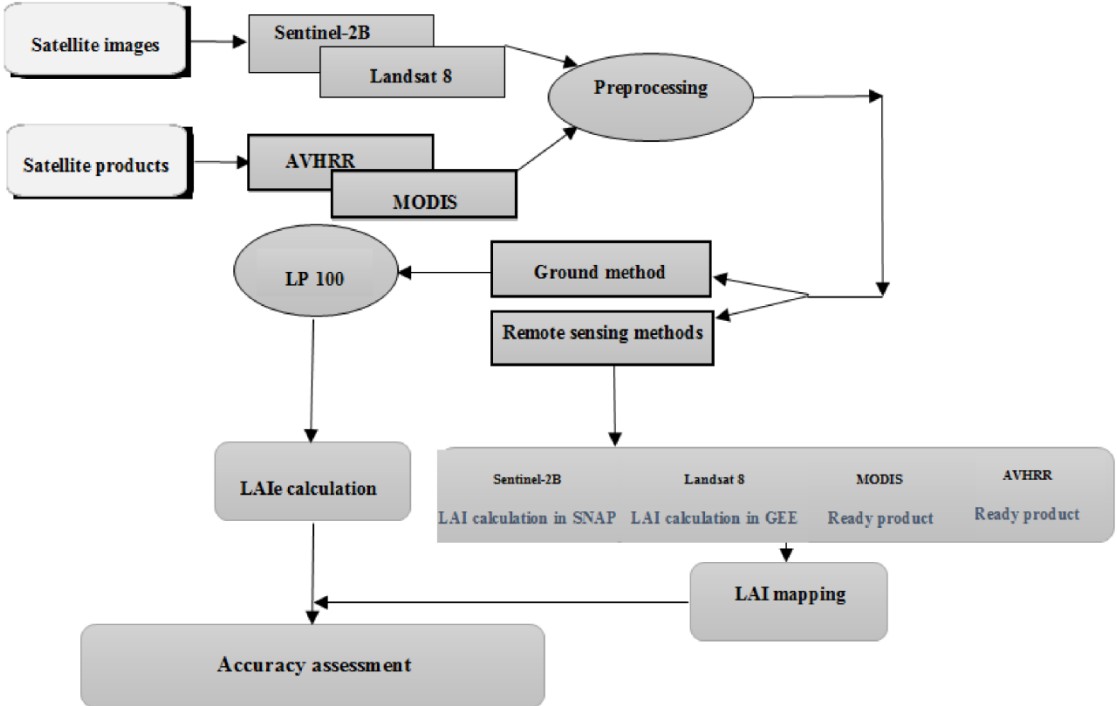

**Figure 3.** Methodological flowchart of the present study.

2.2.3. Statistical Analysis and Validation

For the accuracy assessment, the mean LAI extracted from LP 100 in each homogeneous pixel size of 30 m was compared with the mean LAI extracted from remote sensing. The results were initially evaluated to investigate the degree of agreement between estimated LAI using remote sensing methods and field measurements (LP 100) by calculating the correlation coefficient (r). The Pearson and Spearman tests were respectively used to investigate the correlation state for the normal and non-normal data. The data were examined for normality at a significance level of $p < 0.05$ using the Shapiro–Wilk test (IBM SPSS, Version 26). The LAI data extracted from all sensors in the shrub and Landsat 8 and MODIS in the bush were normally distributed. Besides, the LAI data extracted from Sentinel-2B and AVHRR in the bush and the LAI data extracted from all sensors in the tree were non-normal. The LAI data for June were normally distributed except for MODIS-LAI, and all LAI data extracted from all sensors in July were non-normal.

The coefficient of determination ($R^2$) and five error evaluation criteria were also calculated to achieve the accuracy of the methods used ([38,39]; Equations (5)–(9)). In general, a higher $R^2$ (near one) indicates more accuracy and a lower error [26]. MAE (mean absolute error) (Equation (5)) gives the mean magnitude of estimation errors, and MBE

(mean bias error) (Equation (4)) is the mean estimation error representing the systematic error of an estimation method under or over the LP 100 measurement. RMSE (root mean square error) is also calculated according to Formula (9).

In situations where MAE and MBE are equal or close to zero, it shows that the method used simulates reality well, and by moving away from zero, it shows a small quantity of accuracy or much deviation [40].

$$MAE = \sum_{i=1}^{N} |(LAI_{RS} - LAI_{LP})|/n \tag{5}$$

$$MBE = \sum_{i=1}^{N} (LAI_{RS} - LAI_{LP})/n) \tag{6}$$

MBias (Equation (7)) is expressed as the ratio of LAI remote sensing to the LAI of the LP 100 device. MBias equal to 1, less than 1, and more than 1, respectively, indicate perfect, low, and high estimates. MBias equal to 1 dedicates the closer estimation to the LP 100 values, thereby reflecting the high reliability of the estimates.

$$MBias = \sum_{i=1}^{N} LAI_{RS} / \sum_{i=1}^{N} LAI_{LP} \tag{7}$$

The RBias (Equation (8)) checks for systematic LAI errors obtained using remote sensing methods.

$$RBias = (\sum_{i=1}^{N} (LAI_{RS} - LAI_{LP}) / \sum_{i=1}^{N} (LAI_{LP})) \tag{8}$$

RMSE stands for root mean square error. The RMSE represents the mean of the errors available. It can be used as an essential indicator when our goal is to assess the accuracy of the entire data. Being lower (close to zero) means lower error.

$$RMSE = \sqrt{\sum_{i=1}^{n} \sum_{i=1}^{N} (LAI_{RS} - LAI_{LP})^2 /n} \tag{9}$$

In these equations, n represents the total number of data, $LAI_{RS}$ is the LAI obtained from remote sensing sensors, and $LAI_{LP}$ is the LAI obtained from LP 100.

## 3. Results

The statistical description based on remote sensing methods for different sampling months and different PFTs (shrubs, bushes, and trees) are presented in Tables 3 and 4. In addition, the maps of LAI calculation using different sensors are shown in Figure 4. The scatter plot of sensor-based LAI values according to field-based values for different PFTs and months is respectively shown in Figures 5 and 6. The results of the accuracy assessment for each studied PFT and month are summarized in Table 5 and Figure 7. The results confirmed the spatial and temporal changes of the LAI in Ardabil Province. The lowest LAI has spread from the north to the center and a small part of the province's south.

The lowest and highest mean LAIs obtained in June using Landsat 8 images and Sentinel-2B were 0.67 and 3.13, respectively. In addition, in July, the lowest and highest mean LAIs were estimated to be 0.09 and 3.13, respectively, using Landsat 8 and Sentinel-2B. The lowest and highest mean LAIs based on the PFTs of shrubs using Sentinel-2B were 0.09 and 3.74, respectively. The lowest and highest mean LAIs in bushes using Sentinel-2B, respectively, were 0.21 and 4.40, and in trees using the Sentinel-2B and AVHRR images, they were estimated at 0.3 and 4.40, respectively. The correlation between the LAI obtained from the LP 100 device and the Sentinel-2B images in the shrubs and bushes showed a relatively high correlation coefficient ($|r| > 0.63$) with a corresponding RMSE < 1.37.

**Table 3.** Descriptive statistics of estimated LAI using LP 100 and different sensors extracted for sampling points in different months.

| Months / LAIs | June 2020 | | | July 2020 | | |
|---|---|---|---|---|---|---|
| | Min | Mean | Max | Min | Mean | Max |
| LP 100 | 2.60 | 3.74 | 5.30 | 3.60 | 4.13 | 5.83 |
| Sentinel-2B | 1.53 | 1.92 | 3.13 | 0.09 | 1.24 | 3.13 |
| Landsat 8 | 0.67 | 0.90 | 1.40 | 0.31 | 0.68 | 1.20 |
| MODIS | 0.76 | 1.29 | 2.71 | 0.40 | 0.60 | 1.40 |
| AVHRR | 0.92 | 2.55 | 2.80 | 0.35 | 0.71 | 1.17 |

**Table 4.** Descriptive statistics of estimated LAI using LP 100 and different sensors extracted for sampling points in different PFTs.

| PFTs / LAIs | Shrubs | | | Bushes | | | Trees | | |
|---|---|---|---|---|---|---|---|---|---|
| | Min | Mean | Max | Min | Mean | Max | Min | Mean | Max |
| LP 100 | 0.40 | 2.71 | 4.10 | 2.30 | 5.00 | 6.40 | 2.80 | 4.00 | 6.80 |
| Sentinel-2B | 0.09 | 1.11 | 3.74 | 0.21 | 2.07 | 4.40 | 0.30 | 1.70 | 4.40 |
| Landsat 8 | 0.88 | 0.35 | 1.49 | 0.27 | 0.73 | 1.44 | 1.95 | 0.82 | 0.27 |
| MODIS | 0.20 | 0.99 | 2.13 | 0.29 | 1.14 | 3.43 | 0.70 | 2.40 | 4.30 |
| AVHRR | 0.35 | 1.12 | 2.73 | 0.63 | 1.35 | 3.47 | 0.30 | 0.90 | 2.70 |

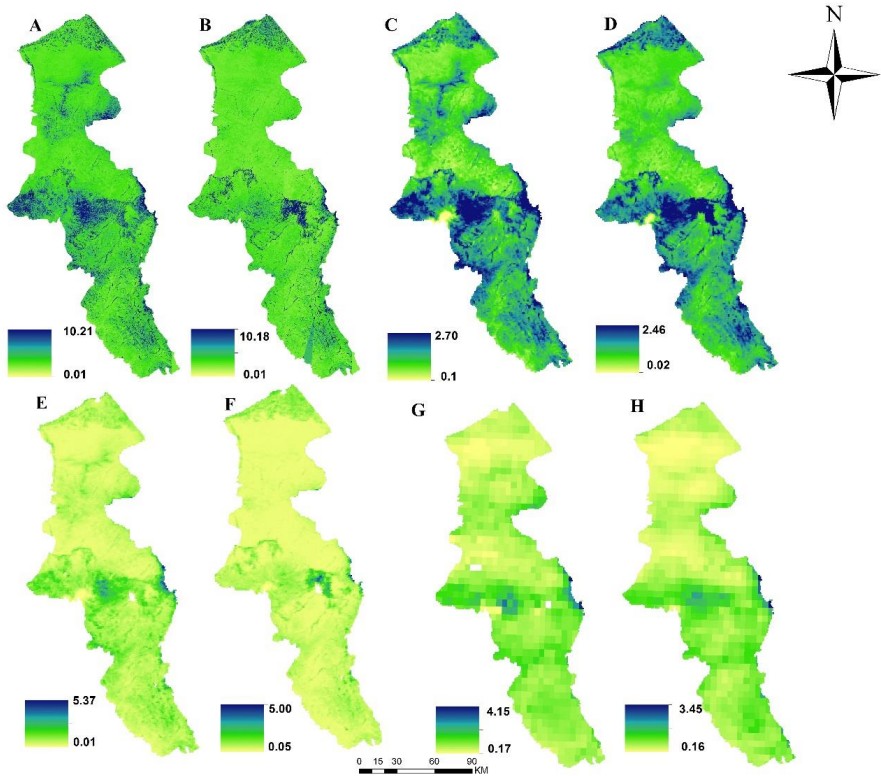

**Figure 4.** LAI estimated respectively in June and July 2020 using Sentinel-2 (**A**,**B**), Landsat 8 (**C**,**D**), MODIS (**E**,**F**), and AVHRR (**G**,**H**).

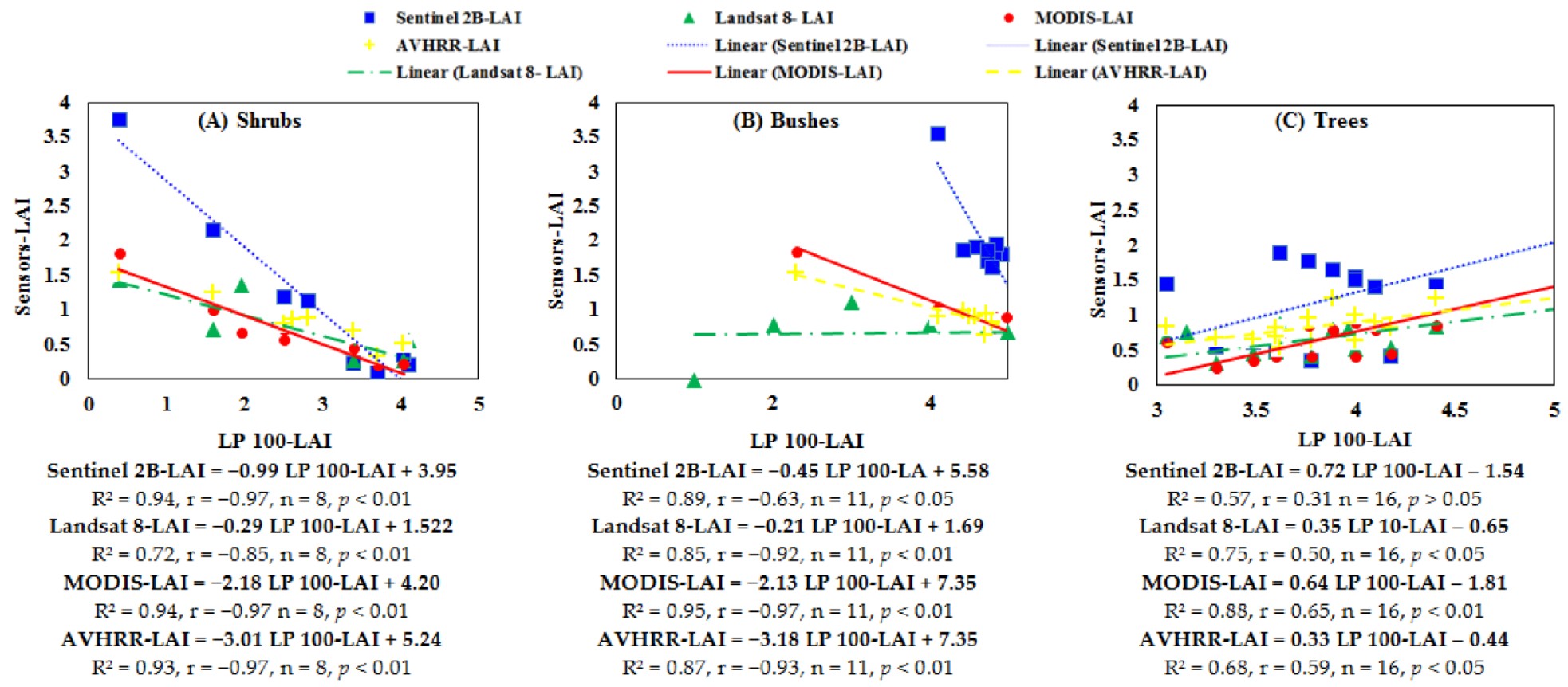

**Figure 5.** Comparison of LP 100 and estimated LAI in different PFTs: (**A**) shrub, (**B**) bush, and (**C**) tree.

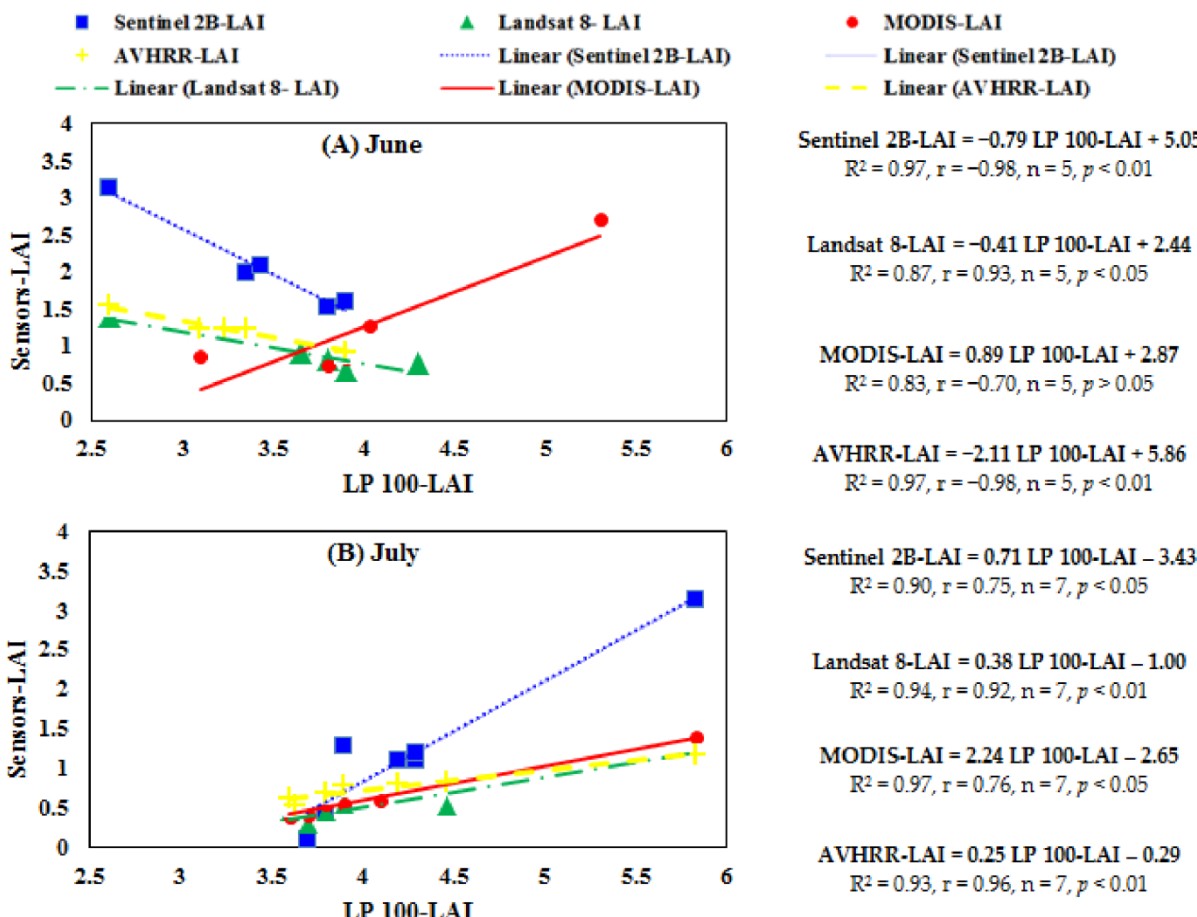

**Figure 6.** Comparison of LP 100 and estimated LAI using Sentinel-2, Landsat 8, MODIS, and AVHRR in June (**A**) and July (**B**) 2020.

**Table 5.** Accuracy assessment results for sampling months, PFTs, and different sensors.

| | Error Statistics | Sensors | MAE | MBE | MBias | RBias | RMSE |
|---|---|---|---|---|---|---|---|
| **Sampling Month** | June | Sentinel-2B | 0.33 | −0.24 | 0.51 | −0.49 | 1.09 |
| | | Landsat 8 | 0.36 | −0.36 | 0.28 | −0.72 | 1.21 |
| | | MODIS | 0.30 | −0.28 | 0.43 | −0.57 | 1.04 |
| | | AVHRR | 0.30 | −0.28 | 0.44 | −0.56 | 1.01 |
| | July | Sentinel-2B | 0.48 | −0.45 | 0.29 | −0.71 | 1.34 |
| | | Landsat 8 | 0.54 | −0.54 | 0.15 | −0.85 | 1.45 |
| | | MODIS | 0.54 | −0.54 | 0.15 | −0.85 | 1.45 |
| | | AVHRR | 0.57 | −0.57 | 0.19 | −0.80 | 1.47 |
| **PFT** | Shrub | Sentinel-2B | 0.23 | −0.16 | 0.36 | −0.64 | 0.86 |
| | | Landsat 8 | 0.20 | −0.18 | 0.27 | −0.73 | 0.77 |
| | | MODIS | 0.20 | −0.16 | 0.34 | −0.66 | 0.78 |
| | | AVHRR | 0.19 | −0.14 | 0.44 | −0.56 | 0.72 |
| | Bush | Sentinel-2 | 0.44 | −0.37 | 0.43 | −0.57 | 1.37 |
| | | Landsat 8 | 0.54 | −0.54 | 0.18 | −0.82 | 1.59 |
| | | MODIS | 0.49 | −0.48 | 0.28 | −0.72 | 1.48 |
| | | AVHRR | 0.48 | −0.05 | 0.29 | −0.71 | 1.45 |
| | Tree | Sentinel-2B | 0.45 | −0.40 | 0.37 | −0.63 | 1.28 |
| | | Landsat 8 | 0.50 | −0.50 | 0.21 | −0.79 | 1.39 |
| | | MODIS | 0.47 | −0.46 | 0.27 | −0.73 | 1.30 |
| | | AVHRR | 0.45 | −0.45 | 0.30 | −0.70 | 1.25 |

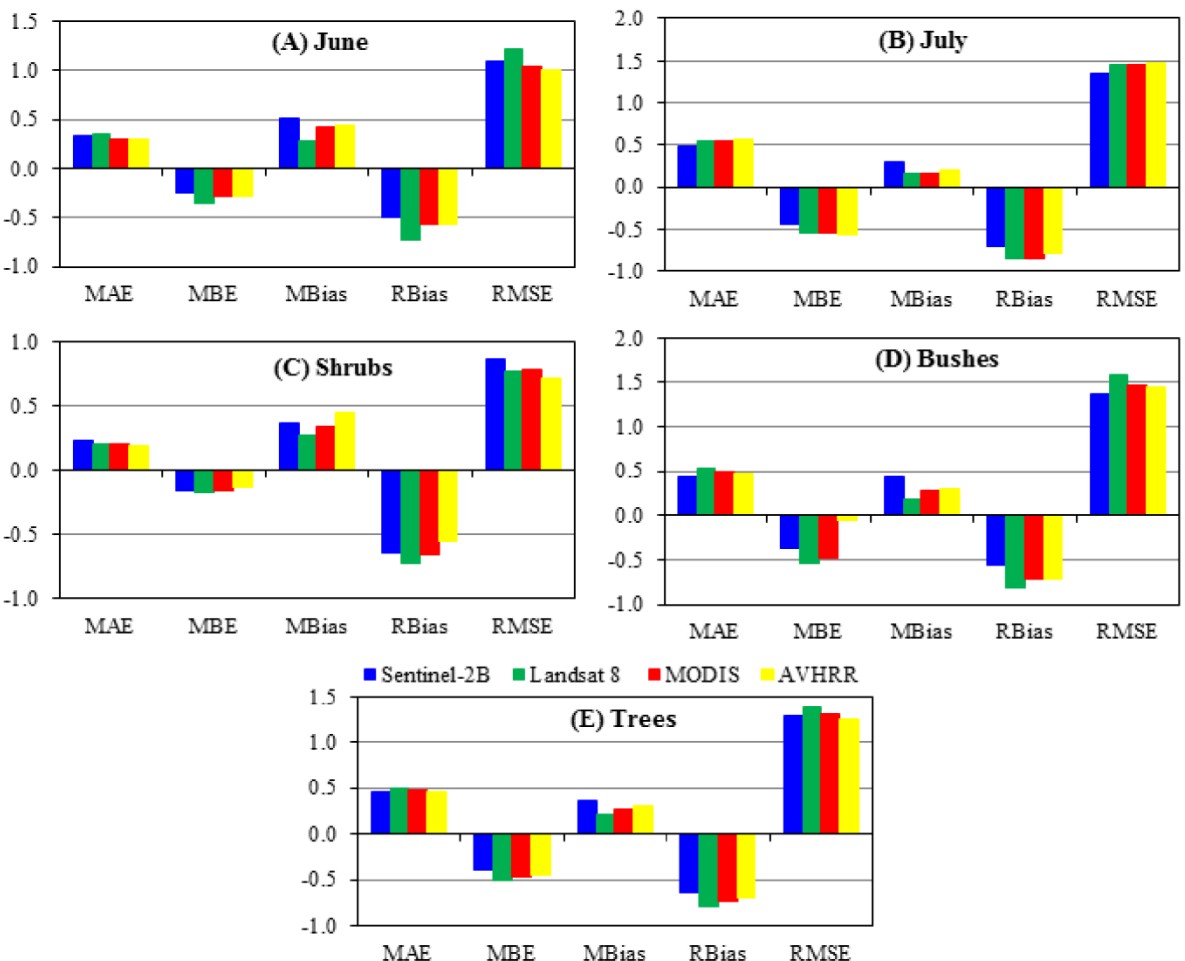

**Figure 7.** Accuracy assessment of studied sensors in different sampling months (**A**,**B**) and PFTs (**C–E**).

## 4. Discussion

Estimation and evaluation of LAI, one of the most important structural characteristics of forest ecosystems, provides much information related to forest dynamics, photosynthesis rate, evaporation and transpiration, net primary production, energy, and carbon exchange coefficient between vegetation and atmosphere [24]. For the estimation of the LAI due to spatial and temporal changes in vegetation canopy, remote sensing methods are low-cost methods for calculating the LAI on a large scale. Despite the high accuracy of direct methods in estimating LAI, they are often time-consuming, costly, and destructive, especially if the level of the studied area is significant. Calculating the LAI based on remote sensing methods and some field methods (e.g., LP 100) is based on the relationship between the LAI and the radiation reflectance characteristics of the canopy measured by sensors. Therefore, sensor-derived data are affected by atmospheric disturbances, sensor characteristics, and sensor accuracy [41–43]. The mean LAI in Landsat 8 was 0.90 in June and 0.68 in July [14]. Overall, in June, the LAI values were estimated to be greater than in July. These results were attributed to the higher levels of vegetation greenness in June, as dedicated by Zhu et al. [44], who reported the special impacts of different seasons on LAI estimation ($p < 0.001$).

The results showed a relatively high correlation between the LAI calculated using the LP 100 device and the remote sensing images in all three vegetation forms (Figure 5). According to the scatter plots in Figure 6, the correlation between LAI values obtained from LP 100 and Sentinel-2B, Landsat 8, and AVHRR images in June, which is the time to examine the forms of shrubs and bushes, was negative. A positive correlation was also reported with the trees collected in July.

Since the pixel value of remote sensing images is averaged from adjacent pixels, it is unlikely to be error-free, so the relationship between them was inversely estimated. The vegetative forms registered in June were bushes and shrubs having leaves smaller than those of trees. Then more light passes through them, and the amount of LAI obtained is less than trees. In general, when the LP 100 device is placed under the leaves of the plants, due to the smaller size and lower layers of the leaves, more light hits the device and is less reflected. However, in trees, the upper layers of the leaves prevent more light from reaching the device. The relationship between the obtained values between tree vegetation forms and remote sensing images was to be reported positive, as trees act like remote sensing images due to larger leaves and less light passing through them. These images estimate the LAI from the top of the crown.

According to the literature [4], measuring the LAI using different tools leads to underestimating about 15–25% of the actual value of the LAI. However, in this study, due to the calibration of the LP 100 device, the accuracy of this device was measured to be higher than the remote sensing methods. There was an underestimation in LAI values in the remote sensing methods compared with LP 100. Myneni et al. [18] reported a nonsignificant correlation between texture variables and effective LAI in evergreen stands through mapping the LAI by linking the spectral, spatial, and temporal information of Landsat 8, IKONOS, and MODIS. Lee et al. [10] also found higher importance of bands in the red-edge and shortwave-infrared than near-infrared bands by comparing the hyperspectral and multispectral data for LAI assessment.

Meyer et al. [45] showed that the vegetation indices created in the infrared bands are more closely related to the LAI. In addition, the prediction models obtained from Landsat 8 data were slightly different from Sentinel-2, and most bands of Sentinel-2B are compatible with Landsat 8 [46]. Sajadi et al. [47] used ETM+, OLI, MODIS, and AVHRR sensors to compare and analyze the NDVI time series. According to the MAE and RMSE, the Landsat 8 sensor had better performance than other selected sensors. In addition, the AVHRR sensor had similar results to Landsat 8, and the MODIS series had lower performance than other sensors in all vegetation classes. Claverie et al. [21] also showed that Sentinel-2B imagery outperformed Landsat 8. Chrysafis et al. [26], in a Mediterranean mixed-forest area in Greece, concluded that the model obtained from the LAI retrieved using Sentinel-2B images and the selection of spectral variables was the most accurate LAI prediction ($R^2$ of 0.85). Liu et al. [24] and Propastin and Erasmi [48] concluded that the LAI values obtained from MODIS images are consistently underestimated. What is more, for wheat LAI retrieval, Yi et al. [49] compared two MODIS land surface reflectance data collections. They found the preferred ability of the 8-day composite data for LAI estimation, thanks to their reduced cloud and aerosol impacts after composting. Therefore, the potential of MODIS data in LAI assessment desires supplementary survey and analysis.

AVHRR was widely used because of its high temporal and moderate spatial ($1.1 \times 1.1$ km) resolutions. Nevertheless, when we use a ready-made AVHRR sensor product, the resolution of this product (LAI_FAPAR/V5′) in estimating the LAI reaches about 5.5 km. Moreover, the prospect of LAI estimation in simple methods necessitating less ground truth points was confirmed by Qi et al. [8] through combining the bidirectional reflectance distribution function (BRDF) model and traditional LAI-VI empirical relation in the AVHRR imagery. Sajadi et al. [47] also indicated that the AVHRR product has a higher temporal resolution. However, its inherent characteristics, such as low spatial resolution, have led to exhibiting noisy behavior in the dataset. Other features, such as design and the significant water vapor absorption due to a broader bandwidth than other sensors, have exhibited noisy behavior in the dataset. MODIS sensor products, including vegetation indices, LAI-photosynthetic radiation fraction, and surface reflection at different spatial and temporal resolutions, have high potential in estimating LAI [49]. Analysis of spectral surface reflectance from Landsat 8 with changes in spatial resolution shows that pixel heterogeneity diminishes at a coarser resolution, and the reflectance is comparable with the MODIS NBAR reflectance product [50].

In remote sensing methods in estimating the LAI, since light has a reciprocal relationship within the crown, satellite data are affected by atmospheric disturbances, the accuracy and specifications of the sensor type, and the process of receiving signals [51]. The relationship between remote sensing and ground-based methods was linear in all cases. In addition, the results showed a positive correlation between them in the trees and a negative one in the shrubs and bushes. Moreover, an MAE of less than 0.54 and an RMSE of less than 1.59 indicated reliable results for the studied months and PFTs. MBE and RBias were calculated for LAI remote sensing (Sentinel-2B, Landsat 8, MODIS, and AVHRR) per month and different PFTs (RBias < −5.60; −0.01 < MBE < −0.57). MBias values between 0.94 and 0.19 indicate an acceptable agreement between remote-sensing- and ground-based measurement data [39]. In addition, MAE less than 0.57 and RMSE less than 1.47 showed reliable results for the studied months and different PFTs. Comparing RMSE among Sentinel-2B, Landsat 8, MODIS, and AVHRR sensors, AVHRR products had a more minor error (0.72) than other images.

## 5. Uncertainties, Limitations, and Future Work

Using satellite data at different temporal and spatial resolutions to estimate LAI may lead to uncertainties and limitations. Spatial scaling issues in the context of validating estimated LAI in this research need minute detail in the future. As Chen et al. [23] noted, the validation of LAI products with different resolutions (moderate: 100–1000 m and coarse: >1 km) is a challenging concern, and they inherently have significant uncertainties owing to the miscellaneous nature of the earth's surface. Comparisons of LAI values from four studied sensors with those aggregated from LP 100 verified the feasibility of LAI deriving, but a few errors still exist. Therefore, developing a model that combines the advantages of both experimental and physical models has a high potential to improve the accuracy of LAI estimation in different temporal–spatial scales for mixed ecosystems. It is noteworthy that the geographic coordinates of the ground area and the pixel may be formally the same. However, the real positions of the ground area may be slightly different due to different approaches to geometric rectification and different geodetic models, sensor peculiarities, solar position, relief, and so on. Besides, pixels from different bands are almost the same, but commonly, they can differ slightly in their exact positions and areas. In the present paper, these uncertainties were not resolved. Due to the large surface area of pixels, slightly more unrealistic values were estimated from sensors, such as Landsat 8 and Sentinel-2B with higher spatial resolution. Fensholt et al. [52] showed that there is around a 2–15% overestimation within MODIS LAI standard products due to a moderate offset unable to be explained by model or input uncertainties. Chen et al. [50] pointed out the 50% to 70% accuracy of AVHRR and SPOT (Satellite Pour l'Observation de la Terre) in LAI estimation due to the surface heterogeneity caused by mixed covers. The bias was mainly due to the uncertainty in the atmospheric correction of Landsat images, but the surface heterogeneity in mixed cover types also caused bias in AVHRR calculations. They attributed the leading cause of random errors to pixels with mixed cover types. Korhonen et al. [20], using band 1 of Sentinel-2B, reported an $R^2$ of 0.73 and an RMSE of 19.60% for boreal forest canopies. They stated that the atmospheric scattering of close pixels could affect the reflection spectrum measured at the field diagram surface; this effect may be more pronounced in heterogeneous landscapes.

In our research, all used images are characterized by moderate or low levels of spatial resolution. At best (Sentinel-2B), a pixel covers an area of about 100 m$^2$ (10 m × 10 m). Such an area may be with several trees, bushes, grass, and forbs. They may represent different taxa of plants. Vegetation composition and plant density may change significantly from one pixel to another. This issue could cause some uncertainties.

The time between the ground data collection and image acquisition is an important factor in estimating the LAI. These findings are not without problems. Moreover, that is why we tried to make the ground data collection time as consistent as possible with the imaging dates. The results of Zheng and Moskal [9] showed that in Canada, the estimated LAI data

matched the ground data, providing better results. The LAI is usually characterized by attributes of the site, stands, and species [53]. In the present research, only the site (different ecoregions) was considered, and other attributes were not investigated. To appropriately interpret the obtained results, it is suggested that the types of species of each PFT would be studied in the future, which has not been performed in this study. Future research could also be continued to implement a suitable method for calculating the LAI in the whole province, which includes describing the land cover types to assess the similarity in the architectural behavior of canopies in different climatic zones.

Examining the efficiency of hyperspectral aerial cameras based on VNIR and SWIR is recommended for further evaluation. In addition, using a suitable sensor of LAI estimation to determine the relationship between LAI and net primary production could be considered for further research. Definitely, using more samples on a larger scale and conducting research in other regions of Ardabil Province, as well as investigating the use of accurate atmospheric corrections and other methods, such as nonlinear or nonparametric regression, can provide the possibility of estimating this important ecological index with greater certainty at the regional level.

## 6. Conclusions

The leaf area index (LAI) is one of the most critical indicators in plant ecology that shows the production capacity of the habitat and its response to environmental changes. Predicting the LAI can be used for various land and vegetation management. The most important advantage of using remote sensing methods is the measurement distance in a short time for the whole province and the ability to repeat and monitor changes. Of course, to determine the accuracy of measurement in these methods, using different direct estimation methods will always maintain its position. We evaluated the utility of Sentinel-2B, Landsat 8, MODIS, and AVHRR for estimating the LAI in forests and rangelands of Ardabil Province, Northwestern Iran. The capability of different images to estimate the LAI and the accuracy of LP 100 as a modern device was investigated.

The mean LAI values extracted by Sentinel-2B, Landsat 8, MODIS, AVHRR, and LP 100 were, respectively, 1.92, 0.90, 1.29, 2.55, and 3.74 for June 2020 and 1.24, 0.68, 0.60, 0.71, and 4.13 for July 2020. All sensors underestimated the LAI in comparison with LP 100. The results of the accuracy assessment criteria showed various results and efficiencies. In terms of different studied months, the lowest MAE, MBE, MBias, RBias, and RMSE were found for MODIS (0.30), Sentinel-2B (−0.24), Landsat 8 (0.28), Sentinel-2B (−0.49), and AVHRR (1.01) in June. Meanwhile, in July, in that respect, the lowest value of statistical errors was found for Sentinel-2B (0.48), Sentinel-2B (−0.45), Landsat 8 (0.15), Sentinel-2B (−0.71), and Sentinel-2B (1.34). Furthermore, among three studied PFTs, the lowest MAE, MBE, MBias, RBias, and RMSE were respectively characterized for AVHRR (0.19) in shrubs, AVHRR (−0.05) in bushes, Landsat 8 (0.18) in bushes, AVHRR (−0.56) in shrubs, and AVHRR (0.72) in shrubs. The achieved results could assist in the efficient selection of proper Sentinel-2B multispectral bands and spectral indices for LAI retrieval in large areas, such as Ardabil Province.

**Author Contributions:** Conceptualization, L.A., A.G. and M.M.; methodology, L.A., A.G. and M.M.; software, L.A., A.G. and F.D.; formal analysis, L.A., A.G. and Z.H.; investigation, L.A., A.G., Z.H. and R.J.; data curation, L.A. and F.D.; writing—original draft preparation, L.A. and A.G.; writing—review and editing, L.A., A.G., Z.H. and R.D.; supervision, A.G. and M.M.; funding acquisition, A.G. and R.D. All authors have read and agreed to the published version of the manuscript.

**Funding:** The research was supported by the Department of Natural Resources, University of Mohaghegh Ardabili, Iran.

**Data Availability Statement:** Available upon request.

**Acknowledgments:** The authors thank and appreciate the University of Mohaghegh Ardabili (Iran), who provided the facilities to do the present study.

**Conflicts of Interest:** The authors have no conflict of interest.

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
