# Peer review of "Multisensor Assessment of Leaf Area Index across Ecoregions of Ardabil Province, Northwestern Iran"

_remotesensing, doi:10.3390/rs14225731_

Round 1

Reviewer 1 Report

This manuscript describes the multi-sensor assessment of LAI in different PFTs. This is a relevant topic making this a valuable manuscript. However, I still have some comments before the manuscript can be accepted.

For assessment of the satellite products’ accuracy (Figure 5), how did the authors ensure that the ground-based LAI measurements matched to satellite-based results with different spatial resolutions? The issue of spatially scaling among four satellite data also should be addressed when LAI measured by LP 100 compared with that derived from satellite data (Figure 6).

Further details of ground-based observations of LAI, including Elementary Sampling Unit, sampling scheme of LP 100, are needed in Section 2.2.1. In addition, the number of LAI measurements obtained using LP 100 should be added.

How did the LAI estimate based on S2 and L8? Please add the retrieval methods or predictive equations in Section 2.2.2.

Line 106, authors will relocate the position of AVHRR (5 Km) and MODIS (500 m) to present satellite data with increasing spatial resolution.

Line 108, the word “first” used here is too absolute. In fact, assessment of LAI retrieved from multi-source satellite data has been widely published.

Line 280, delete (??).

Author Response

Response to Reviewer 1 Comments

This manuscript describes the multi-sensor assessment of LAI in different PFTs. This is a relevant topic making this a valuable manuscript. However, I still have some comments before the manuscript can be accepted.

Response: Thanks to the respected reviewer for the positive attitude towards our work. Authors appreciate the time and effort made by the reviewer in reviewing this manuscript and would like to thank her/him for the constructive comments. All comments and suggestions were incorporated in the revised version by RED Color.

  • For assessment of the satellite products’ accuracy (Figure 5), how did the authors ensure that the ground-based LAI measurements matched to satellite-based results with different spatial resolutions? The issue of spatially scaling among four satellite data also should be addressed when LAI measured by LP 100 compared with that derived from satellite data (Figure 6).

Response: Many thanks to this grateful comment. As you know, the ground-measured data frequently used as references for remote sensing validation (Chen et al., 2002; Wang and Fang, 2020). In addition, ground-based LAI measurement by LP 100 is the only available method for us to check the accuracy of the remote sensing-based LAI. The high efficiency of LP 100 already verified by several references as given below.

  • Photon Systems Instruments. LaiPen LP 100, Manual and User Guide; PSI (Photon Systems Instruments): Drasov, Czech Republic, 2015; p. 45.
  • Cˇerný, J.; Krejza, J.; Pokorny, R.; Bednar, P. LP 100-A new device for estimating forest ecosystem leaf area index compared to the etalon: A methodologic case study. J. For. Sci. 2018, 64, 455–468.

We believe that because we compared the ground data with remote sensing data at the pixel scale, not at the point scale, it could be the correct approach. In addition, the VitiCanopy App results verified the accuracy of LP 100 measurements as mentioned in the paper (r > 0.91; R2 > 0.83; RMSE < 0.51). Anyhow, Chen et al. (2002) in their study noted that the validation of LAI products with different resolutions (moderate: 100 –1000 m, and coarse: >1 km) is a challenging issue and they intrinsically have high uncertainties due to the heterogeneous nature of the Earth’s surface. With this limitation, the ground-measured data still are useful for remote sensing-based LAI map validation. The mentioned uncertainties is added to the manuscript text, too.

“5. Uncertainties, limitations, and future work

Using satellite data at different temporal and spatial resolutions to estimate LAI may lead to some uncertainties and limitations. Spatial scaling issues in the context of validating estimated LAI in this research need minute detail in the future. As Chen et al. (2002) noted that the validation of LAI products with dissimilar resolutions (moderate: 100 –1000 m, and coarse: >1 km) is a challenging concern and they inherently have great uncertainties owing to the miscellaneous nature of the Earth’s surface. Comparisons of LAI values from four study sensors with those aggregated from LP 100 verified the feasibility of LAI deriving, but few errors still exist. Therefore, the development of a model that combines the advantages of both experimental and physical models has a high potential in improving the accuracy of LAI estimation in different temporal-spatial scales for mixed ecosystems.

The LAI is usually characterized by attributes of the site, stands, and species. In the present research, only the site (different ecoregions) was considered and other attributes were not investigated (Emberton, at al,2019). To appropriately interpret the obtained results, it is suggested that the types of species of each PFT would be studied in the future, which has not been done in this study. Future research could be also continued to implement a suitable method for calculating LAI in the whole province, which includes describing the land cover types to assess the similarity in the architectural behavior of canopies in different climatic zones.

Examining the efficiency of hyperspectral aerial cameras based on VNIR and SWIR is recommended for further evaluation. In addition, using a suitable sensor of LAI estimation to determine the relationship between LAI and net primary production could be considered for further research. Definitely, using more samples on a larger scale and conducting research in other regions of Ardabil province, as well as investigating the use of accurate atmospheric corrections and other methods such as non-linear or non-parametric regression, can provide the possibility of estimating this important ecological index with greater certainty at the regional level.”

  • Emberton, S.; Chittka, L.; Cavallaro, A.; Wang, M. Sensor Capability and Atmospheric Correction in Ocean Colour Remote Sensing. Remote Sens. 2016, 8(1), 1. https://doi.org/10.3390/rs8010001
  • Chen, J.M., Pavlic, G., Brown, L., Cihlar, J., Leblanc, S.G., White, H.P., Hall, R.J., Peddle, D.R., King, D.J., Trofymow, J.A., Swift, E., Van Der Sanden, J., Pellikka, P.K.E., 2002. Derivation and validation of Canada-wide coarse-resolution leaf area index maps using high-resolution satellite imagery and ground measurements. Remote Sensing of Environment 80, 165–184. https://doi.org/10.1016/S0034-4257(01)00300-5

  • Further details of ground-based observations of LAI, including Elementary Sampling Unit, sampling scheme of LP 100, are needed in Section 2.2.1. In addition, the number of LAI measurements obtained using LP 100 should be added.

Response: The below explanation was added to the mentioned section for better clarification.

“The field-based observations were recorded using LP 100 device (Figure 2) during June and July 2020. In total, the LP 100 device was applied for 160 shrubs, 117 bushes, and 455 trees throughout different study ecoregions (Table 1). Ecoregions that included only shrubs, bushes, and trees dominance were selected (Figure 1) because of LP 100 limitation for LAI estimation for other PFTs (i.e., grasses, forbs, and dwarfs plant species). The LAI of Hir, Neur, Kowsar, Meshginshahr, and Namin ecoregions were estimated in June 2020 and the others were studied in July 2020. The collected samples were representative of each ecoregion. All recorded values were transferred to the computer system. Then, the Fluor Pen software was applied to obtain the final LAI. The average LAI of those PFTs located in one pixel was compared with the average LAI that estimated by different remote sensing methods of the same pixel.”

Table 1. Sampling number for each plant functional type (PFT) at the study ecoregions

PFT

Number of samples

Ecoregion (sub-ecoregion)

Sampling month in 2020

Shrub

13

Andabil

July

13

Hashtjin (Aghdagh, Berandagh)

July

28

Khalkhal (Isbo, Jafarabad, Majareh, Dilmadeh, Shormineh, Chenarlagh)

July

26

Kowsar (Mashkoul)

June

15

Hatam Meshasi

June

65

Namin Highlands

June

Sum

160 samples

Bush

48

Neur

June

9

Bilesavar-Khoroslo

July

13

Germi

July

13

Andabil

July

10

Hashtjin (Aghdagh, Berandagh)

July

10

Khalkhal (Isbo, Jafarabad, Majareh, Dilmadeh, Shormineh, Chenarlagh)

July

5

Hatam Meshasi

June

9

Namin Highlands

June

Sum

117 samples

Tree

55

Darband Hir

June

10

Neur

June

16

Germi

July

49

Andabil

July

121

Hashtjin (Aghdagh, Berandagh)

July

73

Khalkhal (Isbo, Jafarabad, Majareh, Dilmadeh, Shormineh, Chenarlagh)

July

61

Kowsar (Mashkoul)

June

70

Hatam Meshasi

June

90

Namin Highlands

June

Sum

545 samples

  • How did the LAI estimate based on S2 and L8? Please add the retrieval methods or predictive equations in Section 2.2.2.

Response:  The requested information is added to the manuscript as follows.

“The Snap software was used to estimate the LAI from Sentinel-2B images after calling the images and atmospheric correction in the Sen2core plugin and then resampling it to an image with an accuracy of 10 m. Furtheremore, for the Landsat 8 image, the EVI was selected to calculate the LAI in the GEE system. After calling the Landsat 8 images with clouds less than 45% in Google Earth Engine, the EVI is calculated using the following formula.

)(3)

LAI = 3.618 × EVI - 0.118

(4)

Where, EVI, NIR, RED, and BLUE are respectively indicated the enhanced vegetation index, near-infrared band, red band, and blue band.”

  • Line 106, authors will relocate the position of AVHRR (5 Km) and MODIS (500 m) to present satellite data with increasing spatial resolution.

Response: Thanks for the comment. It was done.

  • Line 108, the word “first” used here is too absolute. In fact, assessment of LAI retrieved from multi-source satellite data has been widely published.

Response: It was accordingly revised.

  • Line 280, delete (??).

Response: It was done.

Reviewer 2 Report

Review

The purpose of this paper is to present an analysis of leaf area index estimation using field and multi-sensor remote sensing datasets in Iran. I think the paper is good, but there are a lot of issues that need to be addressed. The reviewer is concerned about the following:

Abstract

i.                     Line 20-21: ‘the LAI products of GEE were considered’ Are authors referring to the LAI that is available in GEE?

ii.                   ‘Significant correlation (p < 0.05) for all study PFTs between LP 100’. Replaced with ‘studied

iii.                 ‘Because the values of LAI obtained by remote sensing methods are calculated less than field methods, it is better to use when LAI is needed in large-scale studies’. This sentence is not clear to me. Instead, could the authors consider replacing it with this sentence: ‘Despite the fact that remote sensing methods provide less accurate LAI estimates than the field methods, it is better to use when LAI is needed in large-scale studies’

The study area

i.                     ‘Table 1. General characteristics of the selected ecoregions in Ardabil province (adopted from Andalibi et al.s [12])’ I Despite the fact that the Table comes from the same authors, and the manuscript, if accepted, will be published by the same journal, could there be a copyright issue?

Field data collection

i.                     How did the authors integrate the field data in terms of geographical extent per plot considering that their remote sensing datasets are at different spatial resolutions?

ii.                   Line 177-178: Because the viewing angle in the device is completely open (112 degrees on one 176 axis), it is necessary to prevent entering direct light from the Restriction cup. In addition, Paraphrase

Statistical analysis and validation

i.                     It is not clear how the model performance were evaluated? Could the authors clearly explain how many data points were used to assess these agreements between the field and different remote sensing datasets?

ii.                   How many points are used for training as well as for validation?

Discussion

i.                     Line 385: ‘in June, the obtained  results are not consistent with other images’ Paraphrase

ii.                   Given the nature of this type of study (based on approaches and datasets employed), I would recommends to the authors to consider providing a separate section in the discussion section to discuss uncertainties, limitations and future work.

 Conclusion

i.                     Line 468-469: ‘The leaf area index (LAI) is one of the most important indicators in plant ecology that shows the production capacity of the habitat and the response of ‘ecologists’ to environmental changes. Could this be an oversight?

ii.                   It might be helpful for the authors to provide some quantitative results in the conclusion section in order to support their argument.

Author Response

Response to Reviewer 2 Comments

The purpose of this paper is to present an analysis of leaf area index estimation using field and multi-sensor remote sensing datasets in Iran. I think the paper is good, but there are a lot of issues that need to be addressed. The reviewer is concerned about the following:

Response: Authors appreciate the time and effort made by the reviewer in reviewing this manuscript and would like to thank you for the constructive comments. All comments and suggestions were incorporated in the revised version by BLUE Color.

Abstract

  1. i. Line 20-21: ‘the LAI products of GEE were considered’ Are authors referring to the LAI that is available in GEE?

Response: Many thanks to this grateful comment. The detailed explanation of LAI calculation in GEE was added to the Section 2-2. In summary, the ready product of LAI in GEE was not used in this paper. The GEE system only used for process our LAI calculations on the Landsat Images using EVI.

  1. ‘Significant correlation (p < 0.05) for all study PFTs between LP 100’. Replaced with ‘studied’

Response: It was replaced.

iii. ‘Because the values of LAI obtained by remote sensing methods are calculated less than field methods, it is better to use when LAI is needed in large-scale studies’. This sentence is not clear to me. Instead, could the authors consider replacing it with this sentence: ‘Despite the fact that remote sensing methods provide less accurate LAI estimates than the field methods, it is better to use when LAI is needed in large-scale studies’

Response: The suggested sentence is replaced. Thanks for rewriting the sentence.

The study area

  1. ‘Table 1. General characteristics of the selected ecoregions in Ardabil province (adopted from Andalibi et al.s [12])’ I Despite the fact that the Table comes from the same authors, and the manuscript, if accepted, will be published by the same journal, could there be a copyright issue?

Response: The mentioned table is replaced with new table including the sampling information. This new arrangement of the information is set up according the requested information by other reviewers. In this way, the copyright issue considered as well.

Field data collection

  1. How did the authors integrate the field data in terms of geographical extent per plot considering that their remote sensing datasets are at different spatial resolutions?

Response: The below explanation was added to the mentioned section for better clarification.

“The field-based observations were recorded using LP 100 device (Figure 2) during June and July 2020. In total, the LP 100 device was applied for 160 shrubs, 117 bushes, and 455 trees throughout different study ecoregions (Table 1). Ecoregions that included only shrubs, bushes, and trees dominance were selected (Figure 1) because of LP 100 limitation for LAI estimation for other PFTs (i.e., grasses, forbs, and dwarfs plant species). The LAI of Hir, Neur, Kowsar, Meshginshahr, and Namin ecoregions were estimated in June 2020 and the others were studied in July 2020. The collected samples were representative of each ecoregion. All recorded values were transferred to the computer system. Then, the Fluor Pen software was applied to obtain the final LAI. The average LAI of those PFTs located in one pixel was compared with the average LAI that estimated by different remote sensing methods of the same pixel.”

  1. Line 177-178: Because the viewing angle in the device is completely open (112 degrees on one axis), it is necessary to prevent entering direct light from the Restriction cup. In addition, Paraphrase

Response: It was rewriten as follow:

“As the angle of view of LAI sensor is open (112° on the horizontal axis), it is necessary to prevent entering direct light into the restriction cup. In other words, overexposure of an LAI sensor can lead to misinterpretation of actual light conditions. Therefore, before each measurement, it is essential to place the device as shown in Figure 2 to follow the standard principles.”

Statistical analysis and validation

  1. It is not clear how the model performance were evaluated? Could the authors clearly explain how many data points were used to assess these agreements between the field and different remote sensing datasets?

Response: For the accuracy assessment, the mean LAI extracted from LP 100 in each homogeneous pixel size of 30 m were compared with mean LAI extracted from remote sensing.

  1. How many points are used for training as well as for validation?

Response: We didn’t separate the data to two parts of training and validation. Because, the LAI modeing was not the main aim of this research. The comparision between all field data and all remote sensing data was done. This result was enough for us.

Discussion

  1. Line 385: ‘in June, the obtained  results are not consistent with other images’ Paraphrase

Response: Due to the accuracy of Landsat and Sentinel images (30 and 10 meters), the correlation of LAI values with the field method with the values obtained from remote sensing images was positive. Between the two Modis and AVHRR images, the accuracy in the prepared product of LAI of Modis images was higher than that of AVHRR  images, but because Modis images are taken every 4 days and AVHRR  images are daily, that is why there is a negative correlation with the bushes  and  shrubs. Another reason was in the June images, which is most likely due to the presence of clouds in the images and the fact that these products are ready, the correlation was negative with other images compared to the month of July.

  1. Given the nature of this type of study (based on approaches and datasets employed), I would recommends to the authors to consider providing a separate section in the discussion section to discuss uncertainties, limitations and future work.

Response: The below section was added.

“5. Uncertainties, limitations, and future work

Using satellite data at different temporal and spatial resolutions to estimate LAI may lead to some uncertainties and limitations. Spatial scaling issues in the context of validating estimated LAI in this research need minute detail in the future. As Chen et al. (2002) noted that the validation of LAI products with dissimilar resolutions (moderate: 100 –1000 m, and coarse: >1 km) is a challenging concern and they inherently have great uncertainties owing to the miscellaneous nature of the Earth’s surface. Comparisons of LAI values from four study sensors with those aggregated from LP 100 verified the feasibility of LAI deriving, but few errors still exist. Therefore, the development of a model that combines the advantages of both experimental and physical models has a high potential in improving the accuracy of LAI estimation in different temporal-spatial scales for mixed ecosystems.

The LAI is usually characterized by attributes of the site, stands, and species. In the present research, only the site (different ecoregions) was considered and other attributes were not investigated (Nilson, 1999). To appropriately interpret the obtained results, it is suggested that the types of species of each PFT would be studied in the future, which has not been done in this study. Future research could be also continued to implement a suitable method for calculating LAI in the whole province, which includes describing the land cover types to assess the similarity in the architectural behavior of canopies in different climatic zones.

Examining the efficiency of hyperspectral aerial cameras based on VNIR and SWIR is recommended for further evaluation. In addition, using a suitable sensor of LAI estimation to determine the relationship between LAI and net primary production could be considered for further research. Definitely, using more samples on a larger scale and conducting research in other regions of Ardabil province, as well as investigating the use of accurate atmospheric corrections and other methods such as non-linear or non-parametric regression, can provide the possibility of estimating this important ecological index with greater certainty at the regional level.”

 Conclusion

  1. Line 468-469: ‘The leaf area index (LAI) is one of the most important indicators in plant ecology that shows the production capacity of the habitat and the response of ‘ecologists’ to environmental changes. Could this be an oversight?

Response: Sorry, for this unwanted mistake. It was revised.

  1. It might be helpful for the authors to provide some quantitative results in the conclusion section in order to support their argument.

Response: Many thanks to this grateful comment. We provided quantitative results in the conclusion section as given below.

“The results of accuracy assessment criteria showed various results and efficiencies. In terms of different study months, the lowest MAE, MBE, MBias, RBias, and RMSE found for MODIS (0.30), Sentinel-2B (-0.24), Landsat 8 (0.28), Sentinel-2B (-0.49), and AVHRR (1.01), respectively in June. While in July, in that respective the lowest value of statistical errors were found for Sentinel-2B (0.48), Sentinel-2B (-0.45), Landsat 8 (0.15), Sentinel-2B (-0.71), and Sentinel-2B (1.34). Furthermore, among three study PFTs, The lowest MAE, MBE, MBias, RBias, and RMSE were respectively characterized for AVHRR (0.19) in shrub, AVHRR (-0.05) in Bush, Landsat 8 (0.18) in Bush, AVHRR (-0.56) in Shrub, and AVHRR (0.72) in shrub.”

Reviewer 3 Report

Dear colleagues,

Your manuscript is interesting, but there are at least two main and very serious flaws:

(1) Actually a bush and a shrub is synonymous words, both describe one life form of plants. Why do you try to differ them (and how, and for what)?

(2) You used satellite images of Sentinel-2B, Landsat 8, MODIS, and AVHRR. All these images are characterized by moderate or low levels of spatial resolution. At the best (Sentinel), a pixel covers an area about 100 sq m (10 x 10 m). Such area may be with several trees, dozens of bushes and numerous grass and forbs. They may represent quite different taxa of plants. Vegetation composition and plant density may change significantly from one pixel to another. This means satellite images with moderate and low levels of resolution may be used only for general estimation of LAI, but not for the LAI of the different life forms. How can you differ impacts of tree and bush LAI for a pixel? What's about grasses and forbs?

Less significant problems

lines 37-38 - LAI is not play an essential role in ecological processes. Plants play, but an index is not!

Subsection 2.2.3  The Pearson correlation was widely used, but this coefficient is commonly used for date with normal distribution. Did you check normality of your data?

Section 3 - many values are repeated across the text, in tables and pm figures. Authors should avoid numerous copies of the same numbers.

lines 380-382 - both trees and shrubs may have quite different sizes of leaves, especially members of different species. Besides, commonly light can not pass through a leaf, because usually a leaf stops (reflects, utilizes) solar radiation.

Author Response

Response to Reviewer 3 Comments

Dear colleagues,

Your manuscript is interesting, but there are at least two main and very serious flaws:

Response: Thanking you very much for your efforts and valuable energy devoted to review our manuscript. The comments and suggestions considerably improved the quality of the work. All comments and suggestions were incorporated in the revised version by GREEN Color.

1) Actually a bush and a shrub is synonymous words, both describe one life form of plants. Why do you try to differ them (and how, and for what)?

Response: Thank you very much for pointing this out. We separated these two PFTs according to their height and branch growing forms. The woody plant with a height of less than 50 cm and a size of small to medium is considered a shrub. While the woody plant with a height of 50 cm - 7 m has plenty of branches growing from both ground and hard stems. We added these explanation in the text for clarification of the manuscript.

“The woody plant with a height of less than 50 cm and a size of small to medium is considered a shrub. While the woody plant with a height of 50 cm - 7 m has plenty of branches growing from both ground and hard stems. Finally, a woody plant with a height of more than 7 m, a single elongated stem, and few or no branches on its lower part is considered a tree. These perennial PFTs were considered for the target ecoregions.”

2) You used satellite images of Sentinel-2B, Landsat 8, MODIS, and AVHRR. All these images are characterized by moderate or low levels of spatial resolution. At the best (Sentinel), a pixel covers an area about 100 sq m (10 x 10 m). Such area may be with several trees, dozens of bushes and numerous grass and forbs. They may represent quite different taxa of plants. Vegetation composition and plant density may change significantly from one pixel to another. This means satellite images with moderate and low levels of resolution may be used only for general estimation of LAI, but not for the LAI of the different life forms. How can you differ impacts of tree and bush LAI for a pixel? What's about grasses and forbs?

Response: When estimating the LAI in each pixel, the vegetative form of the chosen point was determined, then according to the number of repetitions, the average value of the index was estimated only in the same form of vegetation. Also, in each pixel, according to the number of considered repetitions, it was used to measure the LAI to put together the same vegetative forms. So that in most cases there was a vegetative form in each pixel. It is also worth mentioning that in the studied ecoregions, the vegetative forms were almost similar. Anyhow, we added your comment in the manuscript text for further investigation in future.

Less significant problems

  • lines 37-38 - LAI is not play an essential role in ecological processes. Plants play, but an index is not!

Response:  Thanks for your kind reminder. We have made revisions accordingly.

  • Subsection 2.2.3  The Pearson correlation was widely used, but this coefficient is commonly used for date with normal distribution. Did you check normality of your data?

Response: Yes, the data used for each regression was normally distributed.

  • Section 3 - many values are repeated across the text, in tables and pm figures. Authors should avoid numerous copies of the same numbers.

Response: This section is summarized in terms of values.

  • lines 380-382 - both trees and shrubs may have quite different sizes of leaves, especially members of different species. Besides, commonly light can not pass through a leaf, because usually a leaf stops (reflects, utilizes) solar radiation.

Response: When the device is placed under the leaves of the plants, due to the smaller size of the leaves and the lower layers of the leaves, more light hits the device and is less reflected. But in trees, due to the size of the leaves, the upper layers of the leaves prevent more light from reaching the device

Reviewer 4 Report

This study is important in terms of putting forth the effectiveness and success of different remote sensing platforms on LAI estimation. Here are my comments for authors:

- In literature review, the regression between most important vegetation indices should be added.

- The studies which use SWIR region bands should be added. Some of the example studies are:

- https://ieeexplore.ieee.org/abstract/document/9594823

- https://link.springer.com/article/10.1007/s41870-021-00797-6

- Hyperspectral aerial cameras like HySpex VNIR, SWIR may be also evaluated in the study. (optional)

Author Response

Response to Reviewer 4 Comments

This study is important in terms of putting forth the effectiveness and success of different remote sensing platforms on LAI estimation. Here are my comments for authors:

Response: Thank you for your positive feedback. We also appreciate you for your precious time in reviewing our paper and providing valuable comments. The comments and suggestions considerably improved the quality of the work. All comments and suggestions were incorporated in the revised version by PURPLE Color.

  • In literature review, the regression between most important vegetation indices should be added.
  • The studies which use SWIR region bands should be added. Some of the example studies are:
  • https://ieeexplore.ieee.org/abstract/document/9594823
  • https://link.springer.com/article/10.1007/s41870-021-00797-6
  • Hyperspectral aerial cameras like HySpex VNIR, SWIR may be also evaluated in the study. (optional)

Response: Actually, the regression between the vegetation indices and LAI estimation is discussed in our previous published paper as well (https://doi.org/10.3390/rs13152879.). As you can see, here, the main objective of the paper is to establish a regression between LAI derived from different sensors. However, according to your suggestion, the below explanations including the suggested comments and literatures are added to the introduction section. In addition, regarding your very valuable suggestion about the application of the HySpex aerial camera, our library is not equipped with these devices. Then, we embedded your suggestion in the conclusion section. We will do our best to provide these tools through the legal process to receive financial budgets for purchase in the near future.

 “Another common method of remote sensing-based (indirect) methods in estimating LAI is to establish an experimental relationship between LAI and various vegetation indices extracted from visible near infrared (VNIR) and short wavelength infrared (SWIR) bands, including enhanced vegetation index (EVI), soil-adjusted vegetation index (SAVI), and normalized difference vegetation index (NDVI) [6]. These high multispectral and/or hyperspectral based indices significantly enhanced the quality of monitoring the health natural ecosystems and detecting the changes in vegetation biophysical characteristics (Cimtay et al., 2021; Kumer et al., 2022).”

 “Future research could be also continued to implement a suitable method for calculating LAI in the whole province, which includes describing the land cover types to assess the similarity in the architectural behavior of canopies in different climatic zones. Examining the efficiency of hyperspectral aerial cameras based on VNIR and SWIR is recommended for further evaluation. In addition, using a suitable sensor of LAI estimation to determine the relationship between LAI and net primary production could be considered for further research.”

End

Round 2

Reviewer 2 Report

Solid revision

Author Response

The authors appreciate the time and effort made by the reviewer in reviewing this manuscript and would like to thank you for the constructive comments in the previous version of the manuscript. 

Reviewer 3 Report

Dear colleagues,

I believe after the revision your manuscript became more impressive and interesting for wide audience, however, some problems remain.

The main problem partly remains. How can you compare data for ground areas and data for satellite images? (lines 199–201 and the following analysis) Actually it's very hard to associate exactly a ground area and a pixel, because (1) the geographic coordinates of the ground area and the pixel may be formally the same, but the real positions of the plots may be slightly different (due to different approaches to geometric rectification and different geodetic models); (2) the plot area and the pixel area may be also different (due to sensor peculiarities, solar position, relief etc. — e.g., the nominal size of a pixel of Landsat 7 is 15 x 15 m, but the according ground area (on a plain) is about 17 x 17 m); (3) the pixels from different bands are almost the same, but commonly they can differ slightly in their exact positions and areas. This is why I propose to describe this part of methodology in the very explicit manner.  

line 77 — Nowadays high-resolution satellite images are images with resolution less than 1 m. Images with resolution about dozens of meters may be the medium-resolution ones.

line 290 and so on — Generally speaking, the Pearson correlation may be used for normally distributed data... Did you check normality?

Please, check all situations when en-dashes should be used (e.g., for ranges, between years etc.) (e.g., lines 24–27, 554 and so on).

Besides, there are some problems with English constructions and words, e.g.,

line 3 — Ardabil province > the Ardabil Province

lines 14, 32, 109, 500 — (This) studied > This study

line 36 — many disciplines, including > many disciplines including

line 51 — the LAI in a large > the LAI over a large (or across)

line 126 — the Caspian Sea > Caspian Sea

Please, check your text very carefully again.

Author Response

Response to Reviewer 3 Comments

Dear colleagues,

I believe after the revision your manuscript became more impressive and interesting for wide audience, however, some problems remain.

Response: Authors appreciate the time and effort made by the reviewer in reviewing this manuscript and would like to thank you for the constructive comments. All comments and suggestions were incorporated in the revised version by BLUE Color.

The main problem partly remains. How can you compare data for ground areas and data for satellite images? (lines 199–201 and the following analysis) Actually it's very hard to associate exactly a ground area and a pixel, because (1) the geographic coordinates of the ground area and the pixel may be formally the same, but the real positions of the plots may be slightly different (due to different approaches to geometric rectification and different geodetic models); (2) the plot area and the pixel area may be also different (due to sensor peculiarities, solar position, relief etc. — e.g., the nominal size of a pixel of Landsat 7 is 15 x 15 m, but the according ground area (on a plain) is about 17 x 17 m); (3) the pixels from different bands are almost the same, but commonly they can differ slightly in their exact positions and areas. This is why I propose to describe this part of methodology in the very explicit manner. 

Response: Thank you for this constructive comment. Yes, the honorable referee is absolutely right. One of the problems of research like ours is the same issue mentioned by the respected referee. In this study, in order to overcome this issue, we have conducted ground sampling of three vegetation forms (shrubs, bushes and trees), taking into account the large pixel size of the selected sensors, the selected plants were: firstly, selected in large homogeneous areas, which the size of these areas is much larger than the pixels of the selected sensors. Secondly, the selected plants were also tried to be selected at a distance from each other, so that they were representative of the average of the large area, which were selected. However, in general, the other vegetative forms (grasses and forbs) in the study area, and the limitation of the device used (Laipen LP 100) is still a problem, which has not been possible to cover due to the limitations of this study. At a moment, we do not have any solution to overcome these uncertainties. Toward this, the following explanations were added to Section “2.2.1 Field data collection (LP 100 device): and Section “5. Uncertainties, limitations, and future work” to achieve the points you mentioned. Hope it meets your expectation.

“The ground sampling of three vegetation forms (shrubs, bushes, and trees) was conducted taking into account the large pixel size of the selected sensors. Accordingly, the selected plants were first, selected in large homogeneous areas; the size of these areas is much larger than the pixels of the selected sensors. Secondly, the studied PFTs were also tried to be selected at a distance from each other, so that they were representative of the average of the large area, which was selected. However, in general, the other vegetative forms (grasses and forbs) in the study area, and the limitation of the device used (Laipen LP 100) are still a problem, which has not been possible to cover due to the limitations of this study.”

“It is noteworthy to note that the geographic coordinates of the ground area and the pixel may be formally the same. However, the real positions of the ground area may be slightly different due to different approaches to geometric rectification and different geodetic models, sensor peculiarities, solar position, relief, etc. Besides, pixels from different bands are almost the same, but commonly they can differ slightly in their exact positions and areas. In the present paper, these uncertainties were not resolved.”  

line 77 — Nowadays high-resolution satellite images are images with resolution less than 1 m. Images with resolution about dozens of meters may be the medium-resolution ones.

Response: The sentence is revised accordingly.

line 290 and so on — Generally speaking, the Pearson correlation may be used for normally distributed data... Did you check normality?

Response: Many thanks for this valuable comment. The normality of all data was carefully checked and the essential revisions were done. The below description was added in the Section “Materials and Methods”.

“The Pearson and Spearman tests were respectively used to investigate the correlation state for the normal and non-normal data. The data were examined for normality at a significance level of P <0.05 using Shapiro–Wilk test (IBM SPSS Version 26). LAI data extracted from all sensors in the shrub and Landsat 8 and MODIS in the bush were normally distributed. Besides, LAI extracted from Sentinel-2B and AVHRR in the bush, and LAI data extracted from all sensors in the tree were non-normal. The LAI data for June were normally distributed except for MODIS-LAI, and all LAI data extracted from all sensors in July were non-normal.”

 Please, check all situations when en-dashes should be used (e.g., for ranges, between years etc.) (e.g., lines 24–27, 554 and so on).

Response: It was done.

Besides, there are some problems with English constructions and words, e.g.,

line 3 — Ardabil province > the Ardabil Province

lines 14, 32, 109, 500 — (This) studied > This study

line 36 — many disciplines, including > many disciplines including

line 51 — the LAI in a large > the LAI over a large (or across)

line 126 — the Caspian Sea > Caspian Sea

Please, check your text very carefully again.

Response: Thanks for your kind reminder. The necessary modifications and improvements were made carefully. In addition, we used the revisions suggested by the English Services Center of the University of Mohaghegh Ardabili.

Thank you again for providing such fruitful and constructive comments. Your reviews considerably improved the quality of the work.

End
